# Transcriptomic plasticity of the hypothalamic osmoregulatory control centre of the Arabian dromedary camel

Panjiao Lin [1], Benjamin T. Gillard [1], Audrys G. Pauža [1,2], Fernando A. Iraizoz [1,9], Mahmoud A. Ali[3], Andre S. Mecawi[4], Fatma Z. Djazouli Alim[5], Elena V. Romanova[6], Pamela A. Burger [7], Michael P. Greenwood [1,11], Abdu Adem [8,10,11✉] & David Murphy [1,11✉]

Water conservation is vital for life in the desert. The dromedary camel (*Camelus dromedarius*) produces low volumes of highly concentrated urine, more so when water is scarce, to conserve body water. Two hormones, arginine vasopressin and oxytocin, both produced in the supraoptic nucleus, the core hypothalamic osmoregulatory control centre, are vital for this adaptive process, but the mechanisms that enable the camel supraoptic nucleus to cope with osmotic stress are not known. To investigate the central control of water homeostasis in the camel, we first build three dimensional models of the camel supraoptic nucleus based on the expression of the vasopressin and oxytocin mRNAs in order to facilitate sampling. We then compare the transcriptomes of the supraoptic nucleus under control and water deprived conditions and identified genes that change in expression due to hyperosmotic stress. By comparing camel and rat datasets, we have identified common elements of the water deprivation transcriptomic response network, as well as elements, such as extracellular matrix remodelling and upregulation of angiotensinogen expression, that appear to be unique to the dromedary camel and that may be essential adaptations necessary for life in the desert.

[1] Molecular Neuroendocrinology Research Group, Bristol Medical School: Translational Health Sciences, University of Bristol, Dorothy Hodgkin Building, Bristol, UK. [2] Faculty of Medical and Health Sciences, University of Auckland, Auckland, New Zealand. [3] Department of Pharmacology, College of Medicine & Health Sciences, United Arab Emirates University, Al-Ain, United Arab Emirates. [4] Laboratory of Molecular Neuroendocrinology, Department of Biophysics, Paulista School of Medicine, Federal University of São Paulo, São Paulo, Brazil. [5] University Blida 1, Faculty of Nature and Life Sciences, Department of Biotechnology and Agroecology, Blida, Algeria. [6] Department of Chemistry and the Beckman Institute, University of Illinois Urbana-Champaign, Urbana, IL, USA. [7] Department of Integrative Biology and Evolution, Research Institute of Wildlife Ecology, Vetmeduni Vienna, Vienna, Austria. [8] Department of Pharmacology, College of Medicine & Health Sciences, Khalifa University, Abu Dhabi, United Arab Emirates. [9] Present address: Gene Therapy and Regulation of Gene Expression Program, Centre for Applied Medical Research—CIMA, University of Navarra, Navarra, Spain. [10] Present address: Department of Pharmacology, Khalifa University, Abu Dhabi, United Arab Emirates. [11] These authors contributed equally: Michael P. Greenwood, Abdu Adem, David Murphy. ✉email: abdu.adem@ku.ac.ae; d.murphy@bristol.ac.uk

To live and thrive in arid environments places enormous evolutionary pressure on processes of water conservation. In the desert, free-standing water is sparsely found in isolated oases and rainfed reservoirs, which makes it important for desert animals to minimize water loss[1]. The physiological adaptations of the one-humped Arabian camel (*Camelus dromedarius*) provide a foremost example of how to survive extended periods without drinking in hot, arid desert environments[2,3].

Water loss is extremely well tolerated in the camel; whilst 12% (of the body weight) would be fatal to non-desert mammals due to cardiac failure resulting from circulatory disturbance[4], the camel can survive up to 30%[5]. The mobilisation of water from the metabolism of fat is also thought to help the camel survive periods of water deprivation (WD), although neither hump size, hump adipocyte volume, nor hump lipid content are altered by 23 days of WD[6], suggesting that lipid mobilization occurs elsewhere. Interestingly, ketosis, has not been reported in camels even after starvation[7,8]. However, it is water economy that is particularly vital for survival in the desert, and in the dromedary camel this is achieved by a number of mechanisms, including minimal evaporative cooling (camels rarely sweat), water extraction from undigested food residues, and tolerance of variation in body temperature from 34 °C at night up to 42 °C during the day[9,10], which prevents the loss of around 5 l of water per day through sweating.

At the level of the kidney, the camel produces low volumes of highly concentrated urine as a consequence of highly efficient reabsorption of water[11]. This is mediated by the actions of the antidiuretic hormone arginine vasopressin (AVP) in the kidney to promote water reabsorption in the collecting duct[12] and sodium reabsorption in the thick ascending limb of the loop of Henle[13]. Circulating AVP increases following WD in the camel[11,14], and the kidney response to AVP is known to be highly sensitive in the camel[15]. AVP is made in the hypothalamus by large magnocellular neurones (MCNs) located in the supraoptic nucleus (SON) and released peripherally into the blood circulation from nerve terminals located in the posterior lobe of the pituitary gland, a neuro-vascular interface through which the brain regulates peripheral organs in order to maintain homoeostasis. SON MCNs also produce oxytocin (OXT), a neuropeptide hormone closely related to AVP that, in addition to its well-known roles in parturition, lactation and other reproductive behaviours[16], is thought to have natriuretic activity at the level of the kidney[17,18].

The capacity of the camel for rapid rehydration is equally remarkable. Presented with water, a dehydrated camel can consume up to of 110 l in 10 min[19] and in doing so restore one third of its body weight loss[20]. As such, camels rapidly return to normal hydromineral balance. Such rapid consumption of large quantities of water following WD would be fatal in temperate species.

Whilst the plethora of plastic processes in the hypothalamus that enable the rat to cope with WD are well known[21] and have been documented in detail at the transcriptome level in a number of rat models[22,23], nothing is known about these mechanisms in the dromedary camel. Thus, to investigate the central control of water homoeostasis in the camel, we have performed transcriptome studies on the SON of this species. We cloned and sequenced the camel *AVP* and *OXT* genes and used this information to describe the distribution of neurons expressing *AVP* and *OXT* mRNAs in the SON. This mapping enabled us to perform precise sample derivation for transcriptome analysis. We used RNA sequencing (RNAseq) to catalogue the transcriptomes of camel SON under control and WD conditions. As a consequence of WD, 209 gene transcripts significantly alter their levels of expression. By comparing camel and rat datasets, we have identified common elements of the WD gene response network, as well as elements that appear to be unique to the

dromedary camel and that may be essential for the unique adaptations that enable life in the desert.

## Results

**Experimental groups**. The experimental protocol of this study is summarized in Fig. 1a. We obtained samples from 19 camels divided into 3 groups; controls (water *ad libitum*, $n = 5$), WD (water deprivation for 20 days, $n = 8$) and rehydrated (water deprivation for 20 days followed by water *ad libitum* for three days, $n = 6$). We have shown that WD increased calculated plasma osmolality in these animals, but values return to normal from 12 h of rehydration onwards[24]. Concurrently, circulating AVP levels increased following WD[14,24] and remained higher than control values throughout the rehydration period[21]. The renin-angiotensin system is known to be important for the maintenance of water balance in WD camel[14]. Here we report that circulating angiotensin II (ANG II) levels in our camels were increased significantly by WD (Fig. 1b), then gradually returned to control levels over the course of rehydration (Fig. 1c).

**Mapping of the dromedary camel SON**. We cloned and sequenced the dromedary camel *AVP* (GenBank: OM963135) and *OXT* (GenBank: OM963134) genes in order to design probes for fluorescent in situ hybridisation. We used RNAscope to map hypothalamic AVP and OXT neurones and to define the structure of the dromedary SON (Fig. 2). The nucleus appears as a confined aggregate of large-sized neurons extending rostral-caudally for >4 mm. A distinct topography was observed, consisting of two major subdivisions, named here as the rostral SON (Fig. 2a–j) and caudal SON (Fig. 2f–l). From rostral to caudal, MCNs are clustered (Fig. 2a) in the dorsal position relative to the optic chiasm (OX, the cross of optic tracts), which we characterized as the rostral SON. More caudally, the rostral SON expands dorsolaterally to the distal end of the OX, away from the third ventricle (3V) (Fig. 2b–j). Proceeding caudally, a second condensed MCN population that was characterized as the caudal SON is formed between the OX and the 3V (Fig. 2f). Once the OX is completely detached from the 3V, it is regarded as the optic tract (OT) that no longer is crossed to form the chiasm (Fig. 2g). Continuing caudally, the rostral SON gradually diminishes in size whilst the size of the caudal SON increases and then declines (Fig. 2f–l). The sterically relative locations of these hypothalamic structures are illustrated in Fig. 2m and Supplementary Movie S1. Higher magnification confocal images (Fig. 2n–p) showed that MCNs express high levels of *AVP* or *OXT* mRNAs (Fig. 2o), with only a small population expressing equivalent quantities of both (Fig. 2p).

**Expression of the *AVP* and *OXT* genes by WD and subsequent rehydration**. Based on our RNAscope mapping, we were able to precisely punch both rostral and caudal SONs from control, WD and rehydrated camels for qRT-PCR analysis of gene expression. We looked at the expression of mature *AVP* and *OXT* transcripts as well as the precursor (intron-containing) heteronuclear RNA (hnRNA) transcription products of these genes (*hnAVP* and *hnOXT*) as a proxy measure of transcription. In the rostral SON (Fig. 3), the relative expression of the mature *AVP* mRNA was significantly increased following WD and remained significantly elevated following rehydration compared to controls (Fig. 3a). A similar trend was observed for the *hnAVP* RNA. Both the mature OXT mRNA and *hnOXT* showed a trend of increasing during WD and rehydration but without statistical significance (Fig. 3a). These data were largely replicated in the caudal SON (Fig. 3b) suggesting similar functions for these two anatomically defined SON structures in the camel. Notably, we found barely detectable

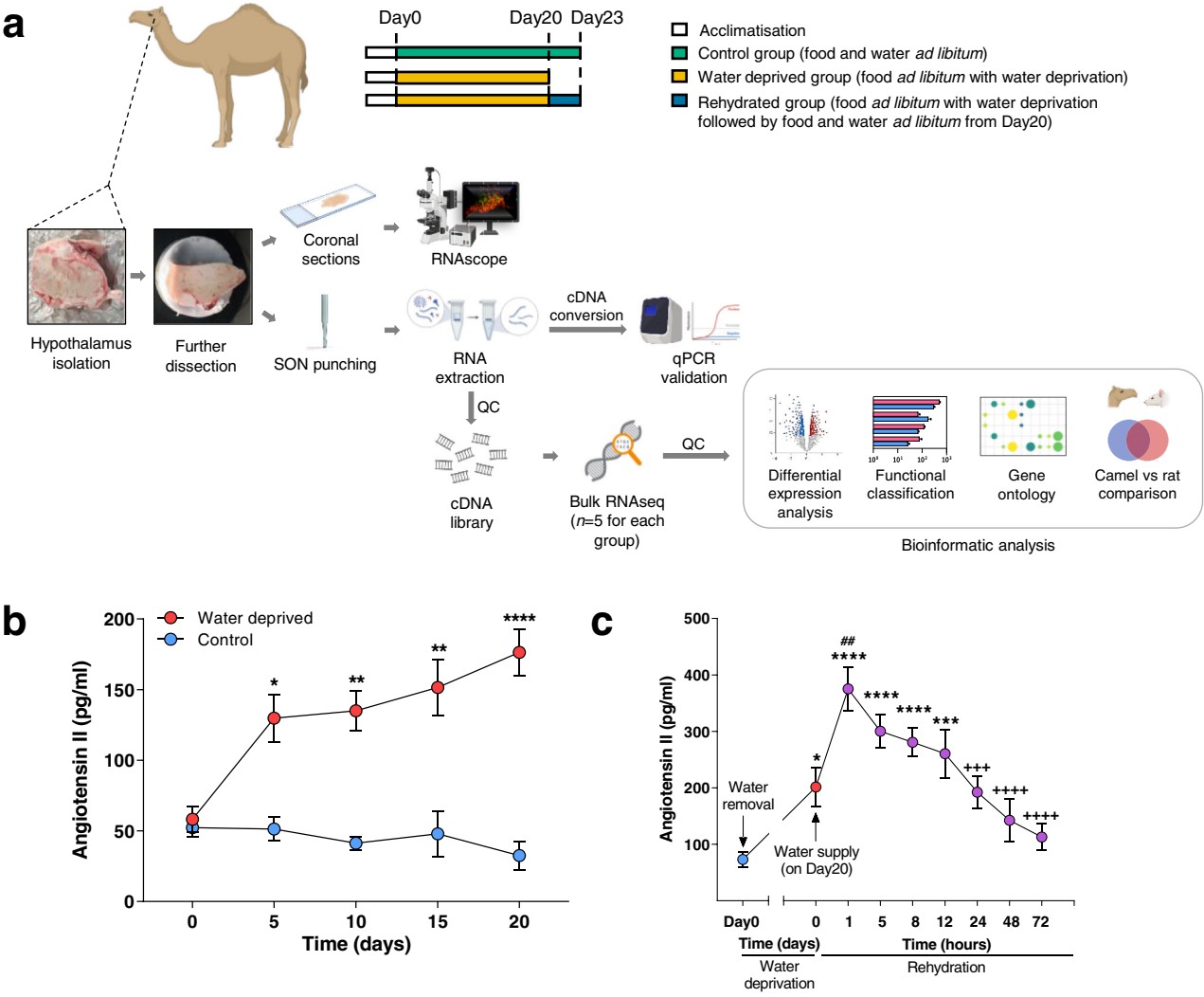

**Fig. 1 Experimental groups and change of plasma angiotensin II level. a** Experimental workflow studying the camel SON (created using BioRender.com incorporating original material by the first author). After acclimatisation, hypothalamic samples were collected from 19 camels divided into 3 groups; control (water *ad libitum*, $n = 5$), WD (water deprivation for 20 days, $n = 8$) and rehydrated (water deprivation for 20 days followed by water *ad libitum* for three days, $n = 6$). A tissue block containing the SON was collected from the camel brain and further dissected to facilitate slicing. Coronal hypothalamic sections were prepared for RNAscope or collection of the SON samples ($n = 5$ for each condition) for RNAseq and/or qRT-PCR. For RNAseq, the population of RNA molecules were converted to cDNA for the next generation sequencing workflow. RNAseq data were analysed using software packages or online databases for quality control (QC), differential expression analysis, functional classification, and gene ontology analysis. The camel SON differentially expressed genes (DEGs) were compared to rat SON DEGs. Plasma ANG II levels during WD and following recovery by rehydration were measured. **b** Plasma ANG II over 20 days of WD compared to control. Data were analysed using two-way repeated measures ANOVA with Šídák's multiple comparisons test. *padj $\leq 0.05$, **padj $\leq 0.01$, ****padj $\leq 0.0001$ in relation to control. **c** Plasma ANG II over 72 h of rehydration (purple dots) for the rehydration camels ($n = 6$) in comparison to their control state (Day0, blue dot) and after 20 days of WD (Day20, red dot). Data were analysed using one-way mixed-effects model (restricted maximum likelihood) for repeated measures with Tukey's multiple comparisons test. *padj $\leq 0.05$, ***padj $\leq 0.001$, ****padj $\leq 0.0001$ in relation to control (Day0). ##padj $\leq 0.01$ in relation to 20 days WD (Day20). +++padj $\leq 0.001$, ++++padj $\leq 0.0001$ in relation to 1 h rehydration.

levels of the *VIP* mRNA, a well-known suprachiasmatic nucleus (SCN) marker gene[25], in both rostral and caudal SONs confirming the identity of the neuroendocrine cells collected as AVP and OXT MCNs (Supplementary Table S1).

**Transcriptome profiles of control and WD camel rostral SON.** We sequenced the ribosomal depleted transcriptomes from the rostral SON of control ($n = 5$) and WD ($n = 5$) camels. Only RNA samples that passed stringent quality control (QC) checks (see Methods) were subject to RNAseq (Supplementary Data S1). A complete profile of basally expressed (control) genes and differentially expressed genes (DEGs) are catalogued in

Supplementary Data S2a. There were 21597 genes expressed in the control samples (average normalized read counts > 0) representing the basal transcriptome of the camel SON (Supplementary Data S2b). This set of genes was classified into functional categories based on the human transcription factor catalogue[26], the pharmacological target database IUPHAR (the International Union of Basic and Clinical Pharmacology)[27] and the physiological function database HGNC (HUGO Gene Nomenclature Committee)[28,29] of their encoded products (Supplementary Fig. S1 and Supplementary Data S3a–i).

We then compared the basal transcriptome to that of the WD state in order to identify DEGs. Principle component analysis

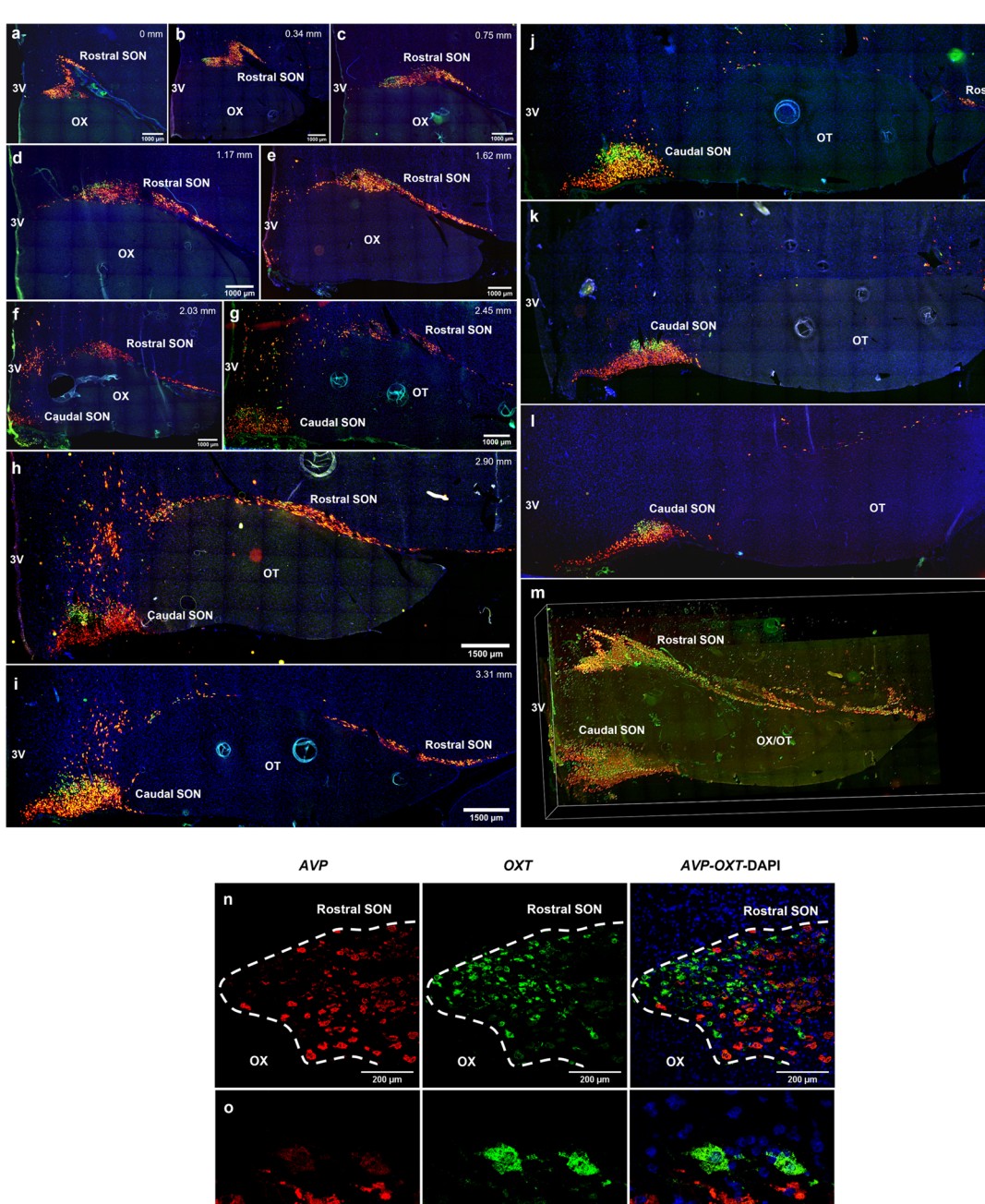

(PCA) showed that the control and WD samples are distinguishable with the rate of 46% on principal component 1 (PC1) and 24% on PC2 of total variance (Fig. 4a). PC1 explained 46% of total variance between the samples and can be attributed to the different hydration conditions.

We tested the validity of using an adjusted *p* value (padj) cut-off of >0.05 for identifying DEGs by using qRT-PCR to look the expression of several genes with RNAseq padj values ranging from 0.062 to 0.15 (Supplementary Fig. S2). This showed that some DEGs were being missed by calling the significance of the DEGs using a padj cut-off of 0.05. However, for the sake of robustness, all our subsequent analyses have been restricted to genes with padj ≤ 0.05, but it is important to note that some genes that failed to reach this cut-off may have biological importance in the transition to the WD state.

**Fig. 2 Mapping of the dromedary camel SON.** The hypothalamic block was sliced into 16 µm thick coronal sections. **a–l** Montage of low power views of RNAscope of *AVP* and *OXT* mRNAs showing their localizations in camel SON. The serial images illustrate the rostral-caudal extent of the SON. Along the rostral-caudal axis, MCNs form a condensed population (rostral SON) lining the dorsal surface of OX (latterly OT), followed up by lateral extension, and form a condensed population (caudal SON) adjacent to the 3V. The level of each section is indicated by the distance from the point that the cell population of rostral SON begins (level 0 mm). **m** A screenshot of Supplementary Movie S1 (a 3D model of camel SON constructed using the RNAscope images) demonstrating spatially relative location of rostral and caudal SON subdivisions. **n** Low power view of RNAscope of *AVP* and *OXT* mRNAs showing their localizations in part of the rostral SON of the same WD camel. **o** Higher magnification of (**n**) distinguishing the AVP and OXT MCNs. **p** Higher magnification of (**n**) highlighting the MCNs with co-localized *AVP* and *OXT* mRNA. *AVP*: red, *OXT*: green, DAPI: blue; SON: supraoptic nucleus, OX: optic chiasm, OT: optic tract, 3V: third ventricle.

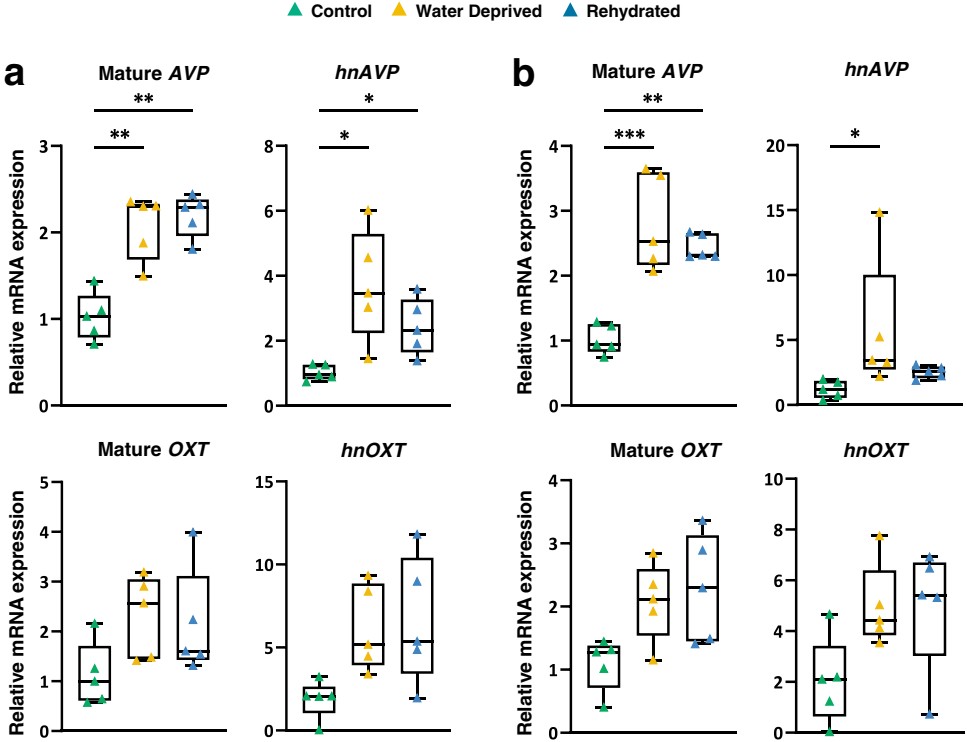

**Fig. 3 Expression of the *AVP* and *OXT* genes in WD and subsequent rehydration.** Relative mRNA expression levels of mature mRNA and heteronuclear RNA of *AVP* and *OXT* in rostral SON (**a**) and caudal SON (**b**) were detected by qRT-PCR. Results are illustrated by box and whisker plots with each individual value as a superimposed triangle (control: green; WD: yellow; rehydrated: blue). Whiskers represent the minimum and maximum values within a group. The median is shown as a line in the centre of the box. Data was analysed by using Brown-Forsythe and Welch one-way ANOVA with Dunnett T3 post-hoc test. *padj ≤ 0.05, **padj ≤ 0.01, ***padj ≤ 0.001.

At a padj cut-off of 0.05, 209 DEGs were identified in camel rostral SON of which 104 were downregulated and 105 were upregulated by WD (Fig. 4b). A volcano plot was used to visualize all the DEGs in WD (Fig. 4c). We next show all upregulated and downregulated DEGs sorted by log2 fold change (LFC) via lollipop charts (Fig. 4d). Amongst these genes, LFC ranged from 3.997 (*VGF*, the most upregulated gene) to −4.664 (*KRT78*, the most downregulated gene). *VGF* is also the most significantly changed gene (padj = 3.43E−29), whilst the most abundantly expressed DEG is *PTPRN* (baseMean = 4068.132). Using the same method for classification of all genes expressed in basal state supplemented with the usage of the UniProt database[30], the 209 DEGs were placed into functional categories (Supplementary Fig. S3 and Supplementary Data S3j).

**qRT-PCR validations of genes changed by WD and subsequent rehydration in camel SON.** We used qRT-PCR to validate DEGs identified by RNAseq in both the rostral and caudal portions of the SON (Fig. 5). In addition, we asked about the effect of rehydration on the expression of these genes. Amongst the genes

examined for RNAseq validation, 13 genes (*AGT*, *ATF4*, *ATP6V0B*, *C1QB*, *CCKAR*, *CREM*, *CTSA*, *FOS*, *PCSK1*, *PDYN*, *PTPRN*, *SCG2* and *VGF*) were upregulated and 3 genes downregulated (*GABBR2*, *COL3A1*, and *CAMK2A*) following WD in the SON. In the rostral SON, we confirmed differential expression of most genes in WD with the only exceptions being *ATF4*, *CAMK2A*, *COL1A1* and *COL3A1* (Fig. 5a). In the caudal SON we confirmed differential expression by WD of genes *AGT*, *ATF4*, *C1QB*, *CCKAR*, *CREM*, *CTSA*, *FOS*, *PCSK1*, *PDYN*, *PTPRN* and *VGF* (Fig. 5b). This perhaps suggests that the anatomical location of the camel SON has some implications for SON function. The recovery of fluid homoeostasis during rehydration prompted the return of most gene transcripts to or towards control levels (Fig. 5). However, there were some notable exceptions. In the rostral SON, the expression during rehydration of *AGT*, *COL3A1*, *PDYN*, *SCG2* and *VGF* remained significantly different from controls. In the caudal SON, the expression during rehydration of *AGT*, *CTSA*, *C1QB*, *PDYN*, *PTPRN* and *VGF* remained significantly different from controls. This suggests that these genes are functionally important during the recovery period following WD.

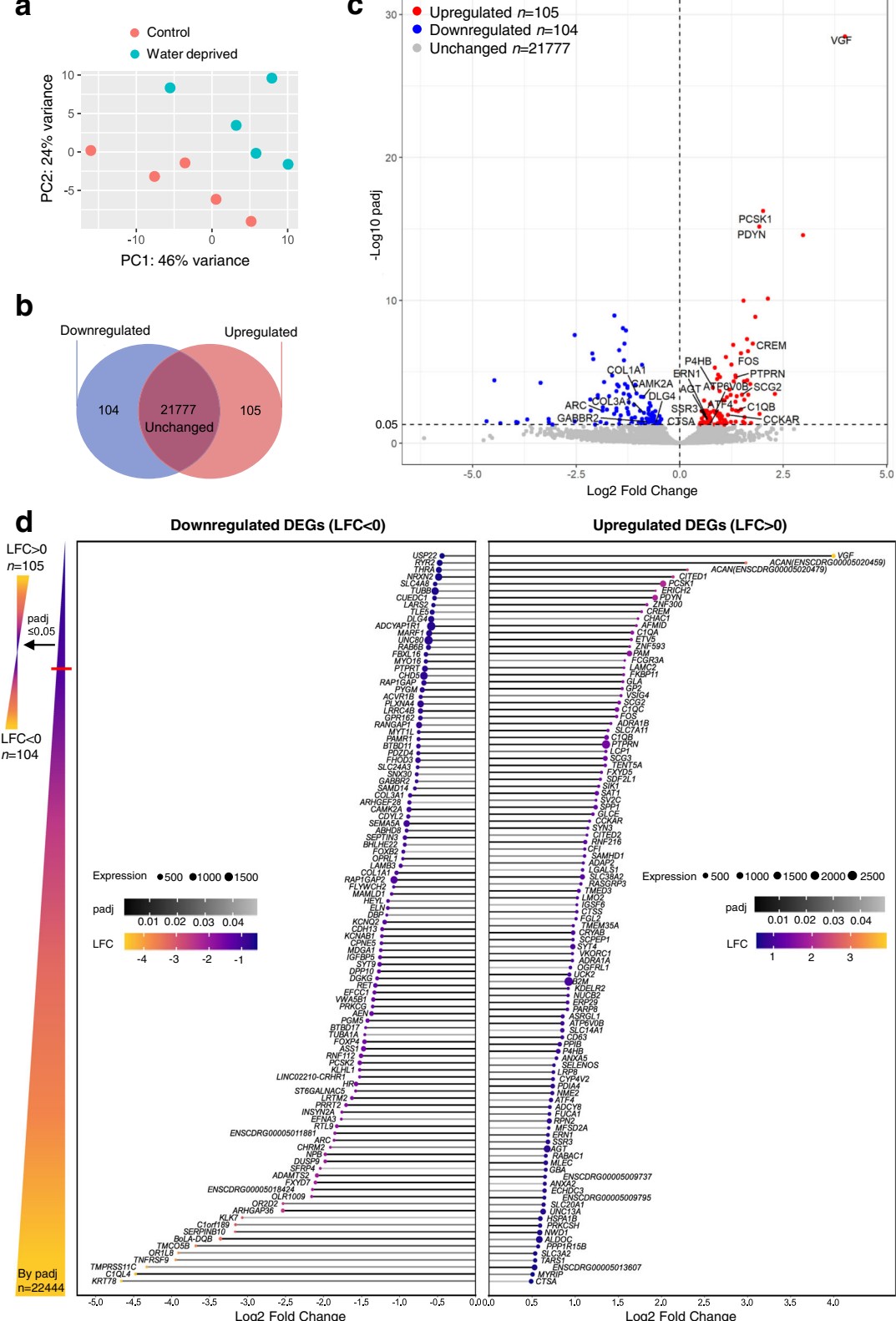

**Fig. 4 Transcriptome profiles of the basal and WD camel rostral SON. a** Principal component analysis (PCA) showing separation between control and WD conditions. Control sample: red; WD sample: turquoise. PC: principal component. PC1 (46%) and PC2 (24%) are the most and second underlying variation between samples. **b** Venn diagram showing 105 upregulated differentially expressed genes (DEGs), 104 downregulated DEGs, and 21777 unchanged genes by WD. **c** Volcano plot of statistical significance (-log$_{10}$ padj) against LFC of DEGs (padj ≤ 0.05) in WD. Red: upregulated DEGs; blue: downregulated DEGs; grey: unchanged genes. Selected DEGs labelled by gene symbols. **d** Upregulated and downregulated DEGs are sorted by LFC. Grayscale of the bars represents padj. Dot size represents transcript abundance measured by average normalized read counts across all samples (baseMean).

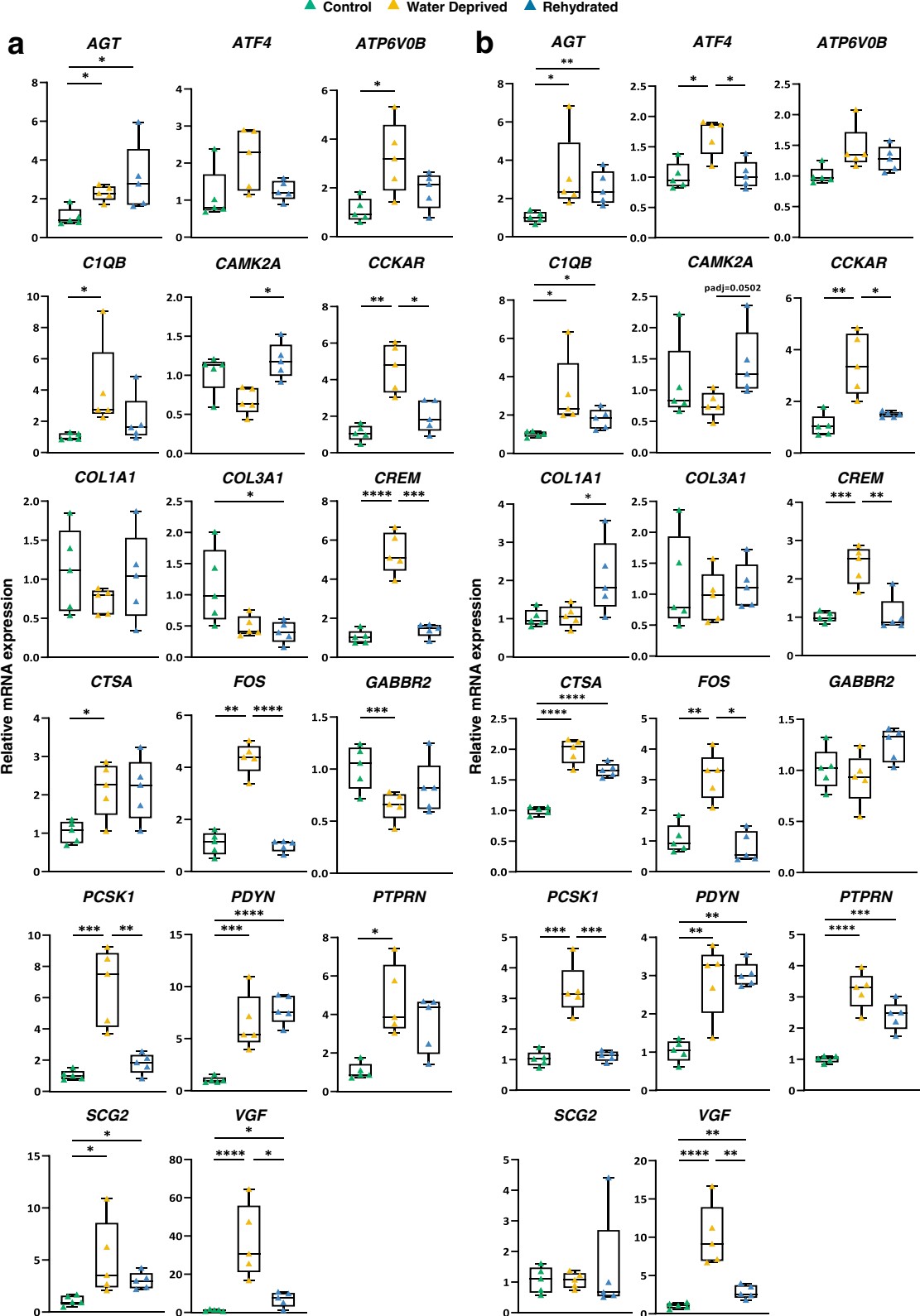

**Fig. 5 qRT-PCR validation of genes changed by WD and effects of rehydration in camel SON.** Differentially expressed genes in WD camel SONs (*AGT, ATF4, ATP6V0B, C1QB, CAMK2A, CCKAR, COL1A1, COL3A1, CREM, CTSA, FOS, GABBR2, PCSK1, PDYN, PTPRN, SCG2* and *VGF*) identified by RNAseq and subjected to qRT-PCR validation. Expression of these genes in both rostral (**a**) and caudal (**b**) SONs. Results (listed in alphabetical order) are illustrated by box and whisker plots with each individual value shown as a superimposed triangle (control: green; WD: yellow; rehydrated: blue). Whiskers represent the minimum and maximum values within a group. The median is shown as a line in the centre of the box. Data was analysed by using Brown-Forsythe and Welch one-way ANOVA with Dunnett T3 post-hoc test. *padj ≤ 0.05, **padj ≤ 0.01, ***padj ≤ 0.001, ****padj ≤ 0.0001.

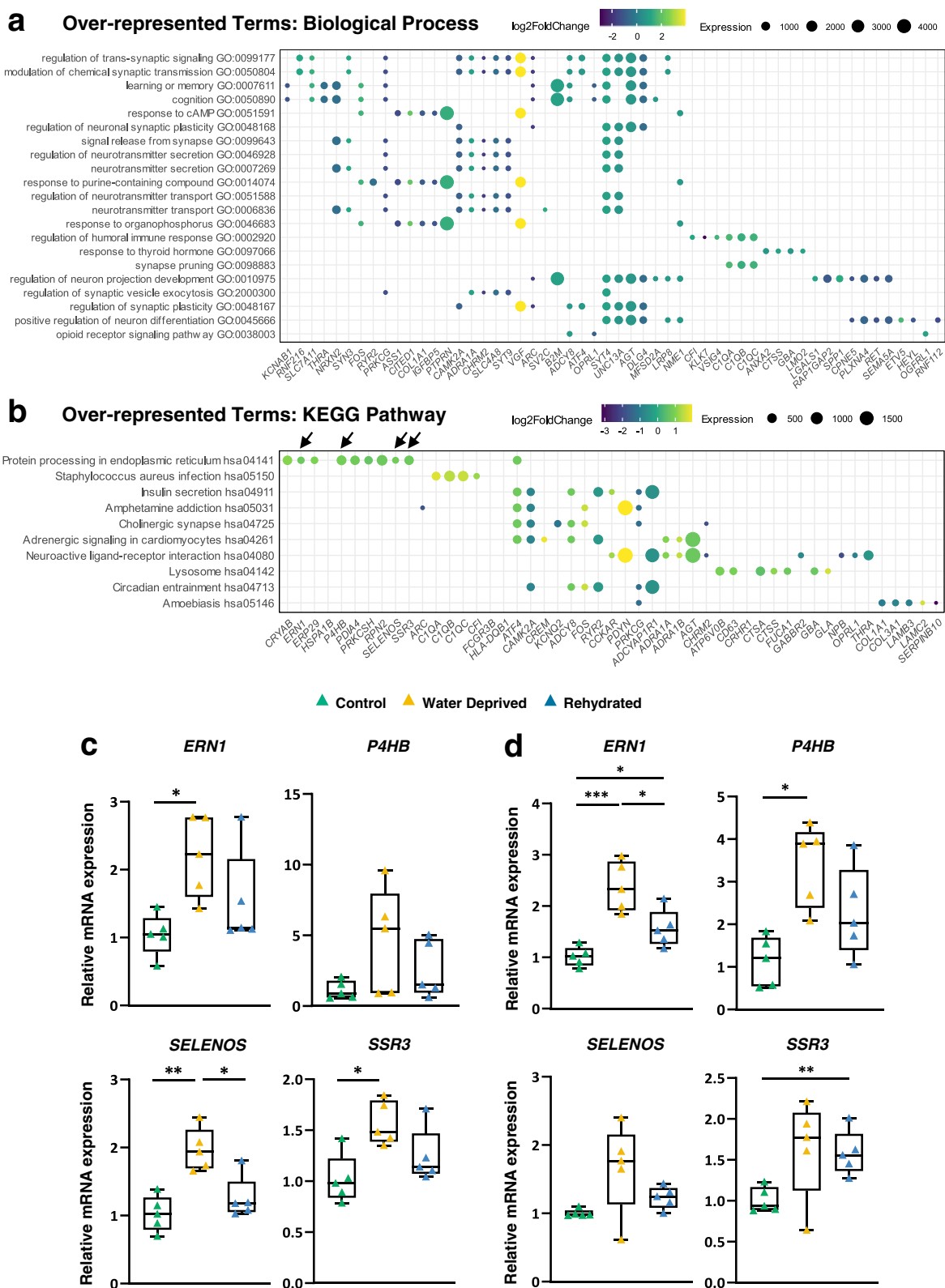

**Gene Ontology of DEGs in the SON of WD camel**. Gene ontology (GO) analysis was performed on SON DEGs identified by RNAseq to characterize gene categories, biological processes and networks which potentially participate in the SON plasticity associated with long-term WD (Fig. 6). The Gene Ontology Biological Processes (GO: BP)[31] and Kyoto Encyclopaedia of Genes and Genomes (KEGG)[32,33] databases were applied for GO

analysis. 21 terms for GO: BP (Fig. 6a) and 10 terms for KEGG pathways (Fig. 6b) were identified to be overrepresented with DEGs regulated by WD (padj ≤ 0.05) using over-representation analysis (Supplementary Data S4a). Amongst the enriched GO: BP terms, 7 terms were related to synapse, 4 were related to neurotransmitter secretion (Fig. 6a). "Modulation of chemical synaptic transmission" (padj=0.0062) and "Regulation of

**Fig. 6 Gene Ontology of DEGs in the WD camel SON.** Over-representation analysis of pathways were performed based on all camel DEGs. **a** Over-represented GO: biological processes. **b** Over-represented GO: KEGG pathways. Benjamini-Hochberg correction (padj ≤ 0.05) was used for multiple comparison correction. Dot plots illustrate the enriched pathways by WD and their associated DEGs. Significantly enriched pathways are listed along the y-axis by padj value from top to bottom in ascending order. Pathway-associated DEGs are denoted by coloured dots. Dot colour and size represent LFC and transcript abundance measured by average normalized read counts aligned to each gene across all samples (baseMean), respectively. Key DEGs [*ERN1*, *P4HB*, *SELENOS* and *SSR3* labelled by arrows in (**b**)] associated to the enriched KEGG term "Protein processing in endoplasmic reticulum" were tested by qRT-PCR in both rostral (**c**) and caudal (**d**) SONs. Results (listed in alphabetical order) are illustrated by box and whisker plots with each individual value as a superimposed triangle (control: green; WD: yellow; rehydrated: blue). Whiskers represent the minimum and maximum values within a group. The median is shown as a line in the centre of the box. Data was analysed by using Brown-Forsythe and Welch one-way ANOVA with Dunnett T3 post-hoc test. *padj ≤ 0.05, **padj ≤ 0.01, ***padj ≤ 0.001.

trans-synaptic signalling" (padj=0.0062) were amongst the most significantly enriched GO: BP terms and were both associated with most camel DEGs (gene counts=18). Notably, the significantly enriched GO: BP term "Response to cAMP" (padj=0.0066, gene counts/term size ratio=8/89) represents a signalling pathway well known to be activated in the SON by WD[34,35]. In the enriched KEGG pathways, "Protein processing in endoplasmic reticulum (ER)" (padj=0.0034) was the most significantly enriched pathway (Fig. 6b and Supplementary Data S4a). Protein Processing in ER is a process that is crucial for protein folding, sorting, and degradation of proteins destined for the secretory pathway[36,37]. The fact that this pathway is activated in WD when protein load on the ER is increased suggests importance in the facilitation of AVP and OXT synthesis and secretion. We visualised this KEGG pathway in Pathview to establish changes to ER function (Supplementary Fig. S4). This revealed differential expression of transcript *ERN1* which encodes inositol-requiring protein 1 (IRE1). Interestingly, IRE1 is one of the major arms of the unfolded protein response (UPR) pathway which triggers intracellular signalling pathways to control ER homoeostasis[38]. Thus, major changes in the WD camel SON centre around modifications to the cell secretory pathway. These observations were validated by qRT-PCR in both the rostral (Fig. 6c) and caudal (Fig. 6d) portions of the SON. The influence of rehydration on the expression of these genes was also studied. Four genes associated with the enriched pathway "Protein processing in ER" were chosen for investigation—*ERN1*, *P4HB*, *SELENOS* and *SSR3*. The upregulated expression by WD of *ERN1, SELENOS* and *SSR3* in the rostral SON, and *ERN1* and *P4HB* in the caudal SON suggests the functional importance of this pathway during WD.

**Comparison of SON transcriptomes between WD camels and rats.** To compare xeric to mesic species in terms of homoeostatic regulation, we compared the camel RNAseq dataset from the present study to our previous RNAseq data of the Wistar rat SON subjected to 72 h of WD[23]. In the rat, the expression levels of 2247 genes are significantly changed (padj ≤ 0.05) by WD (Supplementary Data S2d). Of these, overlap analysis of DEGs revealed 80 common genes, meaning that there are 129 DEGs unique to WD in the camel (Fig. 7a). Of the common DEGs, the majority (70 out of 80) have the same direction of change, whereas 10 (*AEN, BTBD11, CDH13, FHOD3, GLCE, PCSK2, RET, SFRP4, SLC14A1* and *SYT9*) alter expression in the opposite direction (Supplementary Data S2e), which is confirmed by the Spearman correlation test demonstrating a significant ($r = 0.716$, $p < 0.0001$) positive correlation between LFC of the 80 common DEGs in camel and rat (Fig. 7b). To compare the changes in gene expression with the significance level, the Spearman correlation tests revealed significant ($p < 0.0001$) positive correlations between the absolute LFC and -log10padj in both camel ($r = 0.599$) and rat ($r = 0.737$) (Fig. 7c). However, the linear regression revealed a significantly ($p < 0.0001$) smaller slope

in camels (slope=3.962) compared to rats (slope=29.87), demonstrating less variance in the laboratory housed rat samples, compared to the ranch housed camels, as might be expected. By comparing the absolute LFC values of the common DEGs in the two species (Fig. 7d), we found that camels have a significantly ($p < 0.0001$) higher SON transcriptomic response, at least for these 80 evolutionarily conserved transcripts, when compared to rats. The average LFC is almost 3 times higher in camels (1.17) than in rats (0.40), which might be related to the length of stimulus.

We carried out GO analysis on the 80 common DEGs in camel and rat (Fig. 7e, f and Supplementary Data S4b) and the 129 genes (Fig. 7g, h and Supplementary Data S4c) that uniquely changed in camel. The common DEGs were significantly overrepresented by 4 biological processes including "Response to cAMP" (Fig. 7e), and 1 KEGG pathway ("Protein processing in ER") (Fig. 7f) that are overlapped with the pathways overrepresenting all camel DEGs (Fig. 6a, b). We next investigated the 129 DEGs that were unique to the camel. Compared to the pathways overrepresenting all camel DEGs (Fig. 6a, b), the 129 camel-unique DEGs were significantly overrepresented by 8 common biological processes that are primarily related to neurotransmitter, signal release and synaptic plasticity, and 6 different biological processes associated with complement activation, extracellular matrix (ECM) and cell junctions (Fig. 7g). We identified 2 KEGG pathways, "Complement and coagulation cascades" and "ECM-receptor interaction", which are uniquely overrepresented in the 129 unique DEGs (Fig. 7h). The latter KEGG pathway is consistent with structural changes to the ECM in the SON region as a result of long-term WD in the camel[39].

## Discussion
The physiological mechanisms that maintain water homoeostasis have tremendous pressures placed upon them in hot desert environments. We looked to one of nature's water conserving wonders, the camel, to provide answers and molecular insights into the specialisations that ensure survival in these environments. Having recently reported transcriptomic and proteomic adaptations to WD in the camel kidney[40], we turned our attention to the hypothalamic neuroendocrine mechanisms responsible for orchestrating the response to WD. This is the first time that the camel brain has been so extensively studied in terms of its adaptations to the desert environment.

We first comprehensively mapped the structure of the dromedary SON and took forward these anatomical instructions to guide collection of the SON for bulk RNAseq. Mining of transcriptome data by pathways analysis advised us of plastic changes to core cellular processes that are important for secretory and morphological adaptation of the SON. Comparing our new camel SON transcriptome data with that of a mesic species, the rat, has revealed a set of 80 core genes that are commonly regulated in both species, and a set of 129 genes that are uniquely regulated in camel, providing insights into plastic adaptive resilience

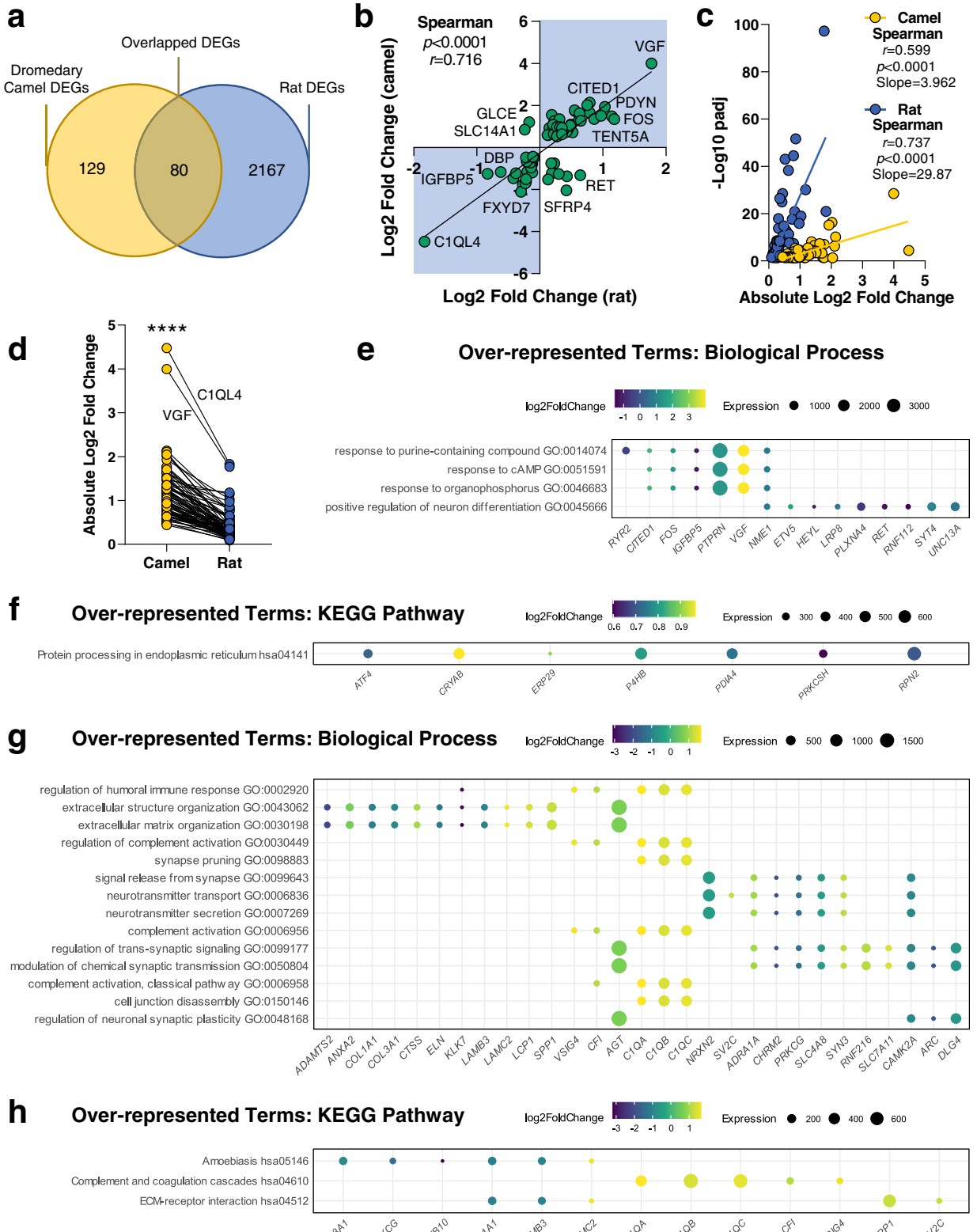

mechanisms to WD that are common in both species or are unique to the camel. However, in comparing these data, we must consider the different protocols employed. Indeed, we note that the 72 h WD protocol in rats evoked a 2.5% increase in plasma osmolality[23], whereas the 20-day WD period in camels resulted in a more severe 13.5% increase in calculated serum osmolality[24].

The molecular details of signalling and transcriptional mechanisms of AVP gene have been studied over decades,

revealing that the cAMP/PKA pathway positively regulates expression of AVP[41]. Indeed, "response to cAMP" was an enriched biological process in the WD camel SON transcriptome, highlighting the importance of the cAMP pathway in defending water balance in both mesic and xeric species.

To defend water balance, the MCNs need to increase protein synthesis to be able to cope with increased demands for neuropeptide secretion. This places pressure on the ER which is the

**Fig. 7 Comparison of SON DEGs in WD camels and rats. a** Venn diagram comparing WD camel and rat DEGs. **b** Simple linear regression and Spearman correlation by LFC of the common DEGs (denoted by dots) in camels and rats. DEGs that are changed in the same direction in expression are highlighted with blue. **c** Simple linear regressions and Spearman correlations between the absolute LFC and -log10padj in camels and rats. **d** Comparison of the absolute LFC values of the common DEGs between camel and rat by Wilcoxon matched-pairs signed-rank test. ****$p < 0.0001$. **e** Over-represented GO: biological processes based on the common DEGs between camel and rat. **f** Over-represented GO: KEGG pathways based on the common DEGs between camel and rat. **g** Over-represented GO: biological processes based on the camel-unique DEGs. **h** Over-represented GO: KEGG pathways based on the camel-unique DEGs. Benjamini-Hochberg correction (padj ≤ 0.05) was used for multiple comparison correction. Dot plots illustrate the enriched pathways by WD and their associated genes. Significantly enriched pathways are listed along the y-axis by padj value from top to bottom in ascending order. Pathway-associated genes are denoted by coloured dots. Dot colour and size represent LFC and transcript abundance measured by average normalized read counts aligned to each gene across all samples (baseMean), respectively.

gateway for proteins entering the secretory pathway. Thus, it is not surprising that the principal enriched KEGG pathway in the WD camel SON was "Protein processing in ER". We have recently reported this pathway as highly regulated component of the rat SON in response to WD[23]. To deliver demands for protein synthesis, AVP neurones also undergo numerous morphological changes including increased size of the perikarya[42,43] and nucleoli[44], and expanded Golgi apparatus[45,46] in humans and rodents. Studies of the rat SON have further described structural changes to astrocytes in WD. The astrocyte processes between magnocellular neurones actively retract and extend during WD to promote the increased release of AVP and OXT[47]. The changes in structural plasticity may be more pronounced in the camel as indicated by alterations to the pathway "ECM-receptor interaction". As DEGs that derive this pathway are uniquely changed in the camel, we suggest that this represents a specialised mechanism of additional structural morphogenesis to improve the supply AVP and OXT in the WD camel. The remarkable seasonal adaptations of the camel SON ultrastructure as shown in our earlier study[39] certainly supports this concept. Future studies will examine the effects of WD on the morphology of camel MCNs.

There are several unique DEGs in the WD camel SON that form the basis for further discussion and investigation. We have shown in the rat SON that the master regulator of the UPR pathway, immunoglobulin heavy chain binding protein (BiP), is increased by WD along with protein kinase RNA-like endoplasmic reticulum kinase (PERK) pathway downstream effectors activating transcription factor 4 (ATF4) and C/EBP-homologous protein[48]. However, this is the first time that alterations to one of the three major ER sensors: PERK, activating transcription factor 6 (ATF6), and IRE1 have been observed in this stimulus. IRE1 is an ER transmembrane sensor that activates the UPR to maintain ER and cellular function. The accumulation of unfolded proteins in the ER activates IRE1α that results in the generation of *XBP1* mRNA by splicing of XBP1u RNA to generate the XBP1s transcript[49–51]. XBP1s promotes the expression of several genes involved in the UPR and ER-associated degradation[52,53]. ER-associated degradation activity is reported to be required by the conformational maturation of AVP prohormone in the ER of AVP-producing neurons[54]. Furthermore, the RNase activity of IRE1 is involved in the regulation of IRE1-dependent decay pathway that degrades mRNAs localised to the ER membrane to reduce the protein load placed on the ER[55]. Increased activation of this pathway may provide further protection to cells in the SON in response to chronic WD. This provides a new mechanism of resilience for the SON in response to high protein loads. Interestingly, MCNs are known to be relatively protected from the aging process in rats[56,57] and humans[58,59] as well as neurodegenerative disease pathologies[58,60]. Thus, changes to the UPR function in camel MCNs represents a future pathway for further exploration of the IRE1 signalling pathway in relation to neuronal cell fate with age and disease.

We also found increased AGT RNA expression only in the WD camel SON. This gene codes for the angiotensinogen (AGT)

protein which when cleaved gives rise to the active ANG II. The MCN expression of AGT and ANG II have been described in the rodent SON, but it is also important to note that AGT is known to be expressed by astrocytes[61]. In the SON, ANG II can promote AVP release[62] by acting on the presynaptic glutamatergic neurones to increase glutamate release[63]. A study performed in mice showed a role for the angiotensin II type 1 A receptors were expressed by AVP MCNs and play a role in the osmotic-induced secretion of AVP[64]. We thus suggest that increasing local angiotensin production in the SON may be an important mechanism in the camel to support AVP secretion in WD. This is coupled with increased peripheral ANG II (Fig. 1b, c) that is known to increase AVP release and elevate blood pressure[18]. Thus, in the camel, WD promotes changes to peripheral and perhaps brain renin-angiotensin systems to instruct AVP release. However, it should be noted that high AGT expression persists following rehydration in both the caudal and rostral portions of the SON, despite decreased circulating ANG II levels. This suggests that SON AGT expression is not contributing the circulating pool, but rather has local functions within the brain. We note that it is a limitation of this study is that we did not measure cardiovascular parameters such as blood pressure and heart rate.

The final gene for general discussion is the gut peptide encoding gene *CCKAR*. The satiety peptide cholecystokinin (CCK) mediates its actions via two G-protein-coupled receptors, CCKAR and CCKBR, and central and peripheral nervous system activation of these receptors inhibits feeding[65]. Interestingly, whilst WD increases *CCKAR* expression in the camels, WD in rats increases expression of the *CCKBR* gene[23], implying cross-species divergence in the regulation of MCNs by the same peptide. Direct administration of CCK into the SON has been shown to increase somatodendritic release of AVP and OXT, and circulating levels of OXT, but not AVP[66]. An additional branch to this network is the synthesis of CCK by MCNs themselves[67,68], which may signal locally to control release of AVP or OXT. Taken together, we reason that SON might be a novel brain region that encode aspects of food/thirst satiety and fluid balance. That said, it is a limitation of this study is that we did not measure food intake.

It should be noted that while the MCNs are the only neuronal cells in the SON, SON is heterogeneous in terms of cell types. In this study, RNAs extracted from the harvested SON samples are from MCNs, and non-neural cell sources including astrocytes, microglia, oligodendroglia, vascular, blood at lower levels[22]. To unequivocally identify the specific transcriptional profiles of different cell types, additional complementary methods such as single cell qPCR[69,70] or single cell RNAseq[71] can be engaged in future studies on the camel SON. A further limitation of this study is that we only studied steady-state transcript levels. Subsequent studies should address the proteomes of the SON and the posterior pituitary gland.

In summary, this comprehensive transcriptomic study provides evidence that the long-term WD induces adaptive changes in the

SON of dromedary camel to defend the animal from prolonged osmotic challenge. We have documented a set of core genes, including *AVP* that encodes AVP, the principal neuroendocrine product of SON and the major regulator of kidney water handling, and core pathways that are commonly changed in WD camel and rat. Same as rat, camel SON may undergo enhanced protein processing that is associated with increased demand of neuropeptide secretion, ER stress and UPR in response to the accumulation of unfolded/misfolded protein during WD. We have confirmed other genes and pathways that are uniquely changed in WD camel and might be indispensable for life in the arid desert, suggesting that camel SON may undergo additional structural remodelling in ECM and upregulation of AGT expression to promote the synthesis and release of AVP and OXT. The enhanced expression of IRE1 specifically taking place in camel supports the concept that UPR functions are actively evoked in the SON as a protective mechanism for the neurons against chronic WD, which prospectively interlinks chronic WD to aging processes and aging-related neurodegenerative diseases. The upregulated *CCKAR* transcription in WD camel SON may assign a novel role to SON to be involved in the regulation of food/thirst satiety in addition to fluid homoeostasis.

## Methods

**Animals**. Samples used in this study were collected from nineteen male dromedary camels aged 4-5 years old, and with a body weight range of 276–416 kg. Body weights for all groups were calculated at baseline date and every five days thereafter using the formula, live weight (Kg) = Shoulder height x chest girth x hump girth x 50[72] as we have reported[73,74]. The camels were supplied with alfalfa hay as feed and were ranch-housed outside Al Ain, United Arab Emirates during the hot months (April and May) of 2016, under careful veterinary supervision. Animals were housed outdoors at ambient temperature. After a short adaptive phase, the camels were divided into a control group ($n = 5$), a WD group ($n = 8$), and a rehydrated group ($n = 6$). The control group had free access to food and water throughout the experimental period. The WD group was supplied with food *ad libitum* but without access to water for 20 days. The rehydrated group was WD for 20 days followed by an unlimited water supply for 3 days.

Blood samples were collected during the experiment by jugular venipuncture. After the experiment, the camels were sacrificed in the local central abattoir for human consumption. The hypothalamus and kidney samples were harvested, frozen on dry ice and shipped frozen on dry ice to the University of Bristol under the auspices of a DEFRA Import Licence (TARP/2016/063) and stored at −80 °C. This study was approved by the Animal Ethics Committee of the United Arab Emirates University (approval ID: AE/15/38) and the University of Bristol Animal Welfare and Ethical Review Board.

**Plasma ANG II assay**. Blood samples were collected between 8:00 and 9:00 am (if not specified) from all groups into heparinized vacutainers, on ice, for hormone plasma measures. Blood from control and WD groups were collected on days 0, 5, 10, 15 and 20 of WD. For the rehydrated group, blood samples were taken on day 0, and at 0, 1, 5, 8, 12, 24, 48 and 72 h following the re-administration of water. Plasma ANG II concentration was determined by specific radioimmunoassay using T-4007 antibody from Peninsula Laboratories, Inc. (San Carlos, CA, USA) as described in a previous study[75]. The ANG II assay sensitivity was 0.39 pg mL$^{-1}$ and the intra- and inter-assay variations were 5.9% and 8.3%, respectively.

**Genomic DNA sequencing**. To clone the dromedary *AVP* and *OXT* genes, genomic DNAs were extracted from camel kidney samples using the DNeasy Blood & Tissue Kit (QIAGEN, 69504), then amplified by PCR (primer sequences available in Supplementary Data S5a) using the Phusion High-Fidelity Master Mix with GC Buffer (Thermo Fisher Scientific, F532L), extracted from the gel using the QIAquick Gel Extraction Kit (QIAGEN, 28704, 28706, 28506 and 28115), A-tailed and ligated into the pGEM®-T Easy Vector Systems (Promega, A1360), and transformed into DH5-alpha competent cells (Thermo Fisher Scientific, 18265017), following the manufacturers' instructions. Vectors harbouring the inserts of interest were harvested using PureYield™ Plasmid Miniprep System Kit (Promega, A1222) and sequenced by Eurofins Genomics using the Sanger dideoxy sequencing method. In order to predict the gene features (i.e., CDS, exons, introns), the obtained sequences were aligned to the published *Camelus ferus AVP* (NCBI: XM_032461957.1) and *Camelus dromedarius OXT* (NCBI: MF464533.1) genes using the Align Sequences Nucleotide BLAST online tool, and also with the dromedary reference genome Camdro2 (GCA_000803125.2)[76]. The sequenced dromedary camel genes with predicted structural features were released to GenBank (*AVP* accession number: OM963135, *OXT* accession number: OM963134).

**Hypothalamic sample processing**. The methodologies for identifying and dissecting the SON from the camel hypothalamus samples are illustrated in Supplementary Fig. S5. Utilizing the 3V and OX as landmarks, the ventral part of the hypothalamus sample containing the SON was dissected (Supplementary Fig. S5a). The rostral and caudal orientations of the SON part were confirmed by recognizing the formation of the OX from OT (Supplementary Fig. S5b) before being mounted to the cryostat sample holder. The brain was sliced into 16 µm thick coronal sections along the rostral-caudal axis using a cryostat set at −20 °C (Leica, CM3050 S). The sections were mounted on Superfrost® Plus slides (Thermo Fisher Scientific, J1800AMNZ) and MCNs of the SON were identified by staining with toluidine blue. This region appeared by eye as a light brown region due to the highly vascularized feature of SON, which was lining the dorsal surface of OX and then lengthening dorsolaterally (Supplementary Fig. S5c). Once identified, sections were collected in a slide box in the cryostat chamber, stained with toluidine blue every 10th slide to trace the journey of the SON. More caudally, a second subregion of MCNs of the SON was identified between the OX and the 3V (Supplementary Fig. S5d). The two subregions of SON were denominated as the rostral SON and the caudal SON. The sections were stored in slide boxes at −80 °C.

**RNA in situ hybridization (RNAscope)**. RNAscope probes for dromedary *AVP* (GenBank: OM963135) and *OXT* (GenBank: OM963134) mRNA were designed by Advanced Cell Diagnostics (ACD) (probe sequences available in Supplementary Data S5b). The mRNA distribution of *AVP* and *OXT* transcripts in the SON of the dromedary camel was analysed by RNAscope Multiplex Fluorescent Assay (ACD, 320851) following the manufacturer's protocol. Briefly, frozen camel brain 16 µm thick sections mounted on slides were fixed in 4% (w v$^{-1}$) paraformaldehyde (PFA) for 15 min on ice, and then immersed in a series of ethanol solutions with increasing concentration (50%, 70%, 100%, 100% v v$^{-1}$), 5 min incubation for each concentration. Slides were air-dried at room temperature (RT) for 5 min before being subjected to the protease IV treatment for 30 min at RT in a humidity control tray. Brain sections were hybridized with probes in humidity control tray at 40 °C for 2 h. Hybridization signals were amplified with RNAscope® Fluorescent Multiplex Detection Reagents (ACD, 320851). DAPI from the RNAscope kit was applied to the brain sections for 45 s at RT. Brain sections were then coated with mounting medium Fluoroshield™ histology mounting medium (Sigma, F6182), coverslipped, and stored at −20 °C. Images were captured with a widefield microscope (Leica, DMI60000) or a confocal (Leica, SP5-II) fluorescent microscope and were analysed using ImageJ (bundled with 64-bit Java 1.8.0_112).

**3D modelling of camel SON**. A 3D reconstruction of the dromedary camel SON was built from RNAscope images. Software including ImageJ (bundled with 64-bit Java 1.8.0_112) and MATLAB (matrix laboratory) were used to process the images. Regarding the landmark structures of hypothalamus (3V and OX), the images were aligned manually using the trakEM2 (blank) function of ImageJ. The brightness and contrast of each channel were also adjusted to increase the signal/noise ratio in ImageJ. The processed images were then firstly scaled down in XY by a factor of 2 using bicubic scaling in ImageJ and then interpolated to aid the visualisation of the 3D data in MATLAB (MATLAB R2018a, version 9.4.0). Briefly, each colour channel was loaded separately, and data was linearly interpolated along the z direction using built-in MATLAB interp3 function to interpolate from the acquired 12 z planes to the desired 96 planes. Data was then saved as individual tiff images, reconstructed to be RGB image, and processed using the 3Dscript plugin of ImageJ to build the 3D model.

**RNA extraction**. Hypothalamic samples were sliced into 100 µm thick coronal sections in a cryostat set at −20 °C (Leica, CM3050 S). The start of the SON was mapped by staining with toluidine blue. SONs were punched from 47 to 60 consecutive slices per sample using a 1 mm micro punch (Fine Science Tools, 18035-01). The punches were dispensed into 1.5 ml tubes maintained on dry ice within the cryostat chamber. At the end of collection, 1 mL of Trizol (Thermo Fisher Scientific, 10296010) and samples were thoroughly mixed by vortexing and stored at −80 °C. Total RNA was extracted using a Direct-zol™ RNA MiniPrep kit (Zymo research, R2052) following the manufacturer's instructions. A Nanodrop spectrophotometer (Thermo Fisher Scientific, ND-1000) was used to determine the RNA concentration.

**RNAseq**. RNA integrity number (RIN) was established for each sample by using the Agilent 2200 TapeStation system (Agilent Technologies, 2503). Control and WD samples ($n = 5$ for each condition) had an average RIN value of 6.27 (range 5.2–7). cDNA libraries were prepared using the TruSeq® Stranded Total RNA Library Prep (Illumina, 15031048) following the manufacturer's instructions. The ribosomal RNA was depleted from the total RNA sample using rRNA removal beads. Following purification, the depleted samples were broken into small fragments (75 bp) that were used for first strand cDNA synthesis and a subsequent second strand cDNA synthesis. The cDNA fragments then went through a single adenine addition and ligation of adapter indices. Library amplifications were performed by PCR to generate the final cDNA libraries. Sample with average cDNA library size close to 260 bp (following the manufacturer's protocol) and higher final library concentration was used for RNAseq. Approximately 200 ng

individual libraries were sequenced using the NextSeq500 High Output Version 2.5, 2 × 75 bp kit (Illumina, 15057931) following the manufacturer's manual. The samples generated averagely 22.2 million mapped reads per sample (range 17.7–37 million). Data was processed using the Real-Time Analysis (RTA) software (version 2.4.6).

**cDNA synthesis and qRT-PCR**. Total RNA (138 ng) was reverse transcribed using the GOScriptTM cDNA synthesis system (Promega, A276A). Primers of target genes (Supplementary Data S5c) were designed based on the reference sequences from National Centre for Biotechnology Information (NCBI)[77], the genome assembly Camdro2 (GCA_000803125.2) and our sequencing data of *AVP* and *OXT*. All primers were synthesized by Sigma-Aldrich®. Intron-specific primers were designed to detect the heteronuclear RNA (hnRNA, pre-mRNA) for *AVP* and *OXT*[78]. The optimization and validation of primers was performed according to ABI protocols and relative standard curve method[79]. cDNA samples were used as templates for the qRT-PCR which was conducted in duplicates in 12 μL reaction volumes using PowerUpTM SYBR Green Master Mix (Thermo Fisher Scientific, 100029283) on an ABI StepOne-Plus Real-Time PCR System. For selecting reference gene of camel SON, the expression level of six commonly used reference genes (*ACTB, B2M, GAPDH, HMBS, HPRT1* and *PPIA*) in rat hypothalamus[80] were checked in our camel RNAseq data. The housekeeping gene *HPRT1* was highly stable in expression under the experimental conditions, thus was selected as the reference gene for the qRT-PCR validation.

**Statistics and reproducibility**. For plasma ANG II measures over the 20 days of WD ($n = 14$, data obtained from the WD treatment in both WD and rehydrated group) in comparison to the control ($n = 5$), data was analysed using two-way repeated measures ANOVA with Šídák's multiple comparisons test in Graphpad Prism (version 9.1.0). Statistics available in Supplementary Data S6a. For plasma ANG II of the rehydrated group ($n = 6$) over 72 h of rehydration in comparison to control and WD states, data was analysed using one-way mixed-effects model (restricted maximum likelihood) for repeated measures with Tukey's multiple comparison test by using Graphpad Prism (version 9.1.0). Statistics available in Supplementary Data S6b.

RNAseq alignment and downstream data analysis were first performed in a Linux-based high-performance computer "Hydra" (PowerEdgeR820 12 core supercomputer; Dell, Round Rock, TX, USA). The paired end sequencing files (FASTQ) were first merged and then trimmed for adaptor sequences using BBDuk tool, followed up by a MultiQC quality check (version 1.9). A dromedary camel reference genome named Camdro2 (GCA_000803125.2) was indexed using Spliced Transcripts Alignment to a Reference (STAR) aligner (version 2.5.3a)[81]. The reads were aligned to the indexed genome. The resulting files were loaded into R (version 4.0.3)[82]. The mapped reads were counted by FeatureCounts[83] where the number of aligned read pairs to each gene for each library were counted. Raw read counts were normalized using Median of ratios method[84] inbuilt in DESeq2[85], DEGs between control and WD ($n = 5$ for each condition) were identified using DESeq2. The inbuilt statistics in DESeq2 was Wald test with Benjamin-Hochberg adjustment[85,86]. Genes with padj ≤ 0.05 were considered to have significant differential expression between groups. Gene annotations were retrieved by using an online tool g: Profiler (g:Convert, Orthology search)[87].

Principle component analysis showing separations between control and WD conditions were plotted by inbuilt function of DESeq2 (ntop=500) based on the regularized log transformed read counts. For comparing different gene sets, unscaled venn diagrams were generated by using an online tool of Bioinformatics & Evolutionary Genomics (http://bioinformatics.psb.ugent.be/webtools/Venn/). Volcano plot showing statistical significance against LFC of genes was generated using ggplot2 package (version 3.3.3)[88] in R. Lollipop chart was generated using ggplot2 to list DEGs by LFC ranking and to deliver information about statistical significance and expression abundance (baseMean: averaged normalized read counts across all samples) of the DEGs.

To compare the common DEGs between WD camel and rat, simple linear regressions and spearman correlation tests were performed using Graphpad Prism (version 9.1.0) on the LFC values of the genes in both species, and the absolute LFC and -log10padj values in each species. The absolute LFC values in the two species were compared by Wilcoxon matched-pairs signed-rank test (two-tailed) via Graphpad Prism (version 9.1.0). Statistics available in Supplementary Data S2f-h.

Gene ontology was performed using ClusterProfiler package (version 3.18.1)[89] in R. For gene ontology of camel, we used the model organism human as references. First, the camel DEG Ensembl IDs were converted to the ortholog human Ensembl IDs by using g: Profiler (g:Orth, Orthology search)[81], then converted to human Entrez IDs using AnnotationDbi package (version 1.52.0)[90] and org.Hs.eg.db package (version 3.12.0)[91], a genome wide annotation database for human, in R. Over-representation analysis[92] was carried out based on the converted gene IDs and the human biological pathway and KEGG pathway annotation databases. Benjamin-Hochberg adjustment[86] was applied for multiple comparison correction to reduce the false discovery rate. Pathways with padj ≤ 0.05 were identified as significantly enriched pathways. Dot plots that visualizing the significantly enriched pathways ranked by padj and the associated DEGs with LFC and abundance (baseMean) in gene expression were plotted using ggplot2 in R.

The $2^{-\Delta\Delta CT}$ method was applied for the relative quantification of gene expression by qRT-PCR[93]. ΔCT is the difference in cycle threshold (CT) values of the gene of interest and the housekeeping gene and was used in statistical tests. ΔΔCT = ΔCT (treated sample) - ΔCT (control sample). $2^{-\Delta\Delta CT}$ was used for plotting. When comparing between control, WD and rehydrated camels ($n = 5$ for each condition), qRT-PCR data was analyzed using Brown-Forsythe and Welch one-way ANOVA with Dunnett T3 post-hoc test in Graphpad Prism (version 9.1.0). Data was illustrated by box and whisker plots to show the dispersion of the dataset. Genes with padj ≤ 0.05 were considered as significantly changed in expression. When comparing between control and WD camels ($n = 5$ for each condition), qRT-PCR data was analyzed using two-way, unpaired t test with Welch correction in Graphpad Prism (version 9.1.0). Genes with $p \leq 0.05$ was considered as significantly changed in expression. Statistics available in Supplementary Data S6.

**Reporting summary**. Further information on research design is available in the Nature Research Reporting Summary linked to this article.

## Data availability
The dromedary camel *AVP* (accession number: OM963135) and *OXT* (accession number: OM963134) gene sequences have deposited to GenBank. The data underlying the transcriptomic analyses, including raw FASTQ files, bulk RNAseq counts, DESeq2-normalized data and project metadata, have been deposited in NCBI's Gene Expression Omnibus (GEO) and are publicly available as of the date of publication (accession number: GSE198577). All software packages used to analyse the data are described in the Methods section, and are common, well-established tools used in omics studies.

## Code availability
This paper does not report original code. All code was adapted from the user manual of the software packages and is available from the corresponding author upon reasonable request. Any additional information required to reanalyze the data reported in this paper is available from the corresponding author upon reasonable request.

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

## Acknowledgements

This research was supported by grants from the Leverhulme Trust (RPG-2017-287) to B.T.G., F.A.I., D.M. and M.P.G., the Biotechnology and Biological Sciences Research Council (BBSRC; BB/R016879/1) to D.M. and M.P.G., the United Arab Emirates University (UAEU)-Program for Advanced Research (UPAR-31M242) to A.A., the São Paulo Research Foundation (FAPESP, 2019/27581-0) to A.S.M., and the UIUC NIDA Centre for Neuroproteomics (P30 DA018310) to E.V.R. Students were supported by grants from the Biotechnology and Biological Sciences Research Council-SWBio DTP programme (BBSRC; BB/M009122/1) to B.T.G. and the British Heart Foundation (BHF; FS/17/60/33474) to A.G.P. We thank Professor José Antunes-Rodrigues (School of Medicine, Ribeirão Preto, São Paulo, Brazil) for the use of laboratory facilities to measure ANG II.

## Author contributions

D.M., A.A., F.Z.D.A., E.V.R. and M.P.G. conceived the project. D.M., M.P.G. and A.A. equally supervised the project. A.A., F.Z.D.A., E.V.R., M.A.A. and M.P.G. performed the animal work and collected the samples. D.M., M.P.G., P.L. and A.A. designed the lab experiments. A.G.P., B.T.G. and P.L. designed the bioinformatic pipeline for RNAseq data analysis. P.L. performed the major laboratory work, bioinformatic analysis (supported by A.G.P., B.T.G. and F.A.I.), data curation, and wrote the manuscript. A.S.M. performed laboratory work and data analysis related to plasma hormone measures, the correlation and linear regression analysis, and absolute LFC comparison between rat and camel. M.A.A. performed laboratory work and data analysis. P.A.B. assembled the reference dromedary camel genome used in this study and provided bioinformatic advice. A.G.P. performed the differential expression analysis of the Wistar rat SON transcriptome used for comparison to the camel transcriptome in this study. All authors contributed to the revision of the manuscript and approved its final version.

## Competing interests

The authors declare no competing interests.
