## [Peer Review File · Communications Biology]

Reviewers' comments:

Reviewer #1 (Remarks to the Author):

In this manuscript entitled "Transcriptomic plasticity of the hypothalamic osmoregulatory control centre of the Arabian dromedary camel" Lin et al has compared the genomic profile of the SON under conditions of water deprivation and rehydration. They identified upregulation of angiotensinogen expression that is unique to the dromedary camels suggesting that this pathway may play an important role in the adaptation of desert life. The manuscript is well written and adds a wealth of data to the field. However, the following may be considered in the analysis and interpretation of the data.

1) I think it would be best if the authors analyze data for genes with a TPM value of 15. It is generally very difficult to confirm the expression by conventional methods when the TPM values are low.

2) The statistical calculations are not clear. For example, I calculated the significance from the raw data for AGT gene TPM values. The mean \pm -SD values are 19804.92 \pm 2610.91 in Control and 19093.62 \pm 890.85 in WD camels with a p value (t-test) of 0.638. It is not clear how the log₂fold change of 0.675417 with a p value of 0.000177 was calculated. The same is true about the calculations of AVP and OXT genes. It will help the reader if the authors describe the statistical analysis in detail.

3) The authors should confirm the major proteins that changed by at least western blotting.

4) The authors should provide the primer sequences for qRT-PCR experiments as supplementary methods.

Reviewer #2 (Remarks to the Author):

The paper is descriptive but contains a lot of very interesting information and the methodology is very sophisticated and rewarding. I have some suggestions

1. It would be ideal to include information on weight change, plasma osmolality, serum Na, plasma copeptin/vasopressin, food intake etc, to better understand the severity of the osmotic challenge. Were they kept in a hot environment or in airconditioned quarters?

2. It would be interesting to know if the dehydrated camels became ketotic as it seems likely that they would have started to breakdown fat as a source of metabolic water.

3. There should be a specific section in the paper focusing on genes known to be regulated by osmotic stress such as aldose reductase, NFAT5/TONEBP, AR-associated target genes, osmotic receptors. There has been a recent paper (J Neurophysiol. 2017 Feb 1;117(2):646-654. doi: 10.1152/jn.00781.2016) that found that Aldose reductase-fructose-xanthine oxidoreductase was stimulated in SON during dehydration in the rat, for example).

4. A lot of interest has focused on inflammation and it would be great to have a paragraph discussing inflammatory related pathways, including inflammatory cytokines, and pathways (inflammasome, NFkB, oxidative stress) as well as mitochondrial pathways. If it is possible to include some histology to see if there is alterations in histology as well.

5. If the authors have other data, such as kidney (especially medulla) or data from fat (lipase) or liver, it could be especially interesting.

"Transcriptomic plasticity of the hypothalamic osmoregulatory control centre of the Arabian dromedary camel"

Reviewer #1:

In this manuscript entitled "Transcriptomic plasticity of the hypothalamic osmoregulatory control centre of the Arabian dromedary camel" Lin et al has compared the genomic profile of the SON under conditions of water deprivation and rehydration. They identified upregulation of angiotensinogen expression that is unique to the dromedary camels suggesting that this pathway may play an important role in the adaptation of desert life. The manuscript is well written and adds a wealth of data to the field. However, the following may be considered in the analysis and interpretation of the data.

1) I think it would be best if the authors analyze data for genes with a TPM value of 15. It is generally very difficult to confirm the expression by conventional methods when the TPM values are low.

We understand the reviewer's point of view. However, we do not want to miss any information in this once in a lifetime dataset. With techniques such as fluorescent *in situ* hybridisation (RNAscope) it has become possible to detect a very small number of transcripts per cell. From another perspective, genes with low TPM values could be abundant in only a small population of cells present in this heterogenous SON brain sample. In summary, we think it is important that nothing is missed from this non-model organism dataset so no arbitrary cut-offs were used.

2) The statistical calculations are not clear. For example, I calculated the significance from the raw data for AGT gene TPM values. The mean \pm SD values are 19804.92 \pm 2610.91 in Control and 19093.62 \pm 890.85 in WD camels with a p value (t-test) of 0.638. It is not clear how the log₂fold change of 0.675417 with a p value of 0.000177 was calculated. The same is true about the calculations of AVP and OXT genes. It will help the reader if the authors describe the statistical analysis in detail.

It would appear that the referee has used the Wistar rat data (Part d of Supplementary Data S2) for their calculations, **not** the camel data (Parts a-c of Supplementary Data S2). The referee has thus confirmed that AGT does not change in rat, although it does in camel. The mean \pm SD values for the camel data are 1414.61 \pm 342.17 in control and 2265.17 \pm 354.53 in WD camels; differences are significant.

We refer the reviewer to the Method section - Statistics and reproducibility for the statistical analyses performed on this data (page 17, line 12). The inbuilt statistics in DESeq2 are Wald test with Benjamin-Hochberg adjustment as multiple test correction. For more details about Wald test, please refer to the DESeq2 vignette:

<http://bioconductor.org/packages/devel/bioc/vignettes/DESeq2/inst/doc/DESeq2.html>

3) The authors should confirm the major proteins that changed by at least western blotting.

The SON is a tiny brain region, and our primary goal was to describe the camel SON transcriptome. Unfortunately, we do not have sufficient camel SON samples to carry out western blot analyses. This is a limitation of working with non-model organisms like the camel.

4) The authors should provide the primer sequences for qRT-PCR experiments as supplementary methods.

All of this information is to be found in Supplementary Data S5.

Reviewer #2:

The paper is descriptive but contains a lot of very interesting information and the methodology is very sophisticated and rewarding. I have some suggestions

1. It would be ideal to include information on weight change, plasma osmolality, serum Na, plasma copeptin/vasopressin, food intake etc, to better understand the severity of the osmotic challenge.

All of these data, derived from the same animals used in this study, are presented in separate manuscripts (references 24, 73, 74) and that are appropriately cited in the text of this manuscript.

Were they kept in a hot environment or in airconditioned quarters?

As we describe (page 12, line 32), the animals were ranch housed. We have added the following explanatory statement: Animals were housed outdoors at ambient temperature.

2. It would be interesting to know if the dehydrated camels became ketotic as it seems likely that they would have started to breakdown fat as a source of metabolic water.

This is an interesting question that we have now addressed in the Introduction (page 3, lines 10-13):

The mobilisation of water from the metabolism of fat is also thought to help the camel survive periods of water deprivation (WD), although neither hump size, hump adipocyte volume, nor hump lipid content were altered by 23 days of WD (reference 6), suggesting that lipid mobilization occurs elsewhere. Interestingly, ketosis, has not been reported in camels even after starvation (reference 7, 8).

3. There should be a specific section in the paper focusing on genes known to be regulated by osmotic stress such as aldose reductase, NFAT5/TONEBP, AR-associated target genes, osmotic receptors. There has been a recent paper (J Neurophysiol. 2017 Feb 1;117(2):646-654. doi: 10.1152/jn.00781.2016) that found that Aldose reductase-fructose-xanthine oxidoreductase was stimulated in SON during dehydration in the rat, for example).

Our RNAseq data suggests that neither aldose reductase (ALR1B1 in camel SON), NFAT5 nor TRPV1 are altered in expression in camel following dehydration. We have performed comparisons with rat SON data which lists genes that are commonly regulated by dehydration in these models. These can be found in Part e of Supplementary Data S2. For brevity and comprehension, the discussion was limited to key findings.

4. A lot of interest has focused on inflammation and it would be great to have a paragraph discussing inflammatory related pathways, including inflammatory cytokines, and pathways (inflammasome, NFkB, oxidative stress) as well as mitochondrial pathways.

We used unbiased ontology analysis to interrogate the functional networks in our datasets. This did not reveal any enrichment for inflammatory or mitochondrial pathways. We did, however, reveal KEGG pathway related to "protein processing in the ER", and our validation experiments and discussion focussed on these genes.

If it is possible to include some histology to see if there is alterations in histology as well.

At the same time as the samples, were taken for this study, tissue was also taken for morphological analysis from the same animals. We refer the reviewer to the point above regarding the limited amount of material available for this complex study.

5. If the authors have other data, such as kidney (especially medulla) or data from fat (lipase) or liver, it could be especially interesting.

We have recently published transcriptome data from camel kidney cortex and medulla (reference 40). We have now cited this manuscript (page 9, line 19-20).

Note that when the hypothalamic samples for this study were collected, all other organs and tissues from these animals were taken at the same time. These samples are currently being studied by numerous groups around the globe. Thus, over the coming years, we will present a comprehensive picture of the overall physiological and molecular responses of the camel to dehydration and subsequent rehydration.

Reviewers' comments:

Reviewer #1 (Remarks to the Author):

No More comments

Reviewer #2 (Remarks to the Author):

I thank the authors for addressing my questions. All in all, the responses were helpful.

I still think that it is important to understand the severity of the dehydration process, and a table summarizing the changes in weight, serum Na, serum osmolality, serum vasopressin or copeptin, food intake, etc, is really important even if the data has been published in other journals. It is sort of key to understanding the nature of the insult, especially given the multiple ways camels adapt to heat and dehydration. The authors also do not describe (if I read the paper correctly) how the dehydration was done in the rats, and whether it was acute or chronic and again the severity of the dehydration using similar parameters. This is important from the standpoint of interpreting differences in response between species.

My guess is that this study is looking at both water and food deprivation in the camel, as spontaneous food intake will likely drop when no water is provided. This might be expected to activate AMPK and likely angiotensin II more as a mechanism to help maintain BP. Was BP measured? Perhaps this should be viewed as a limitation, that this is a study of both energy and water depletion.

Finally, another limitation of the study should be mentioned, which is that most of the measurements are in RNA, and it does not necessarily reflect changes in protein.

Finally, I think that the authors should also comment on the general histology (PAS or H/E) of the SON in the different states to better understand if there are histologic changes.

All in all, however, the paper is very interesting and the authors are congratulated.

COMMSBIO-22-1472A

"Transcriptomic plasticity of the hypothalamic osmoregulatory control centre of the Arabian dromedary camel"

Reviewer #2:

I thank the authors for addressing my questions. All in all, the responses were helpful.

I still think that it is important to understand the severity of the dehydration process, and a table summarizing the changes in weight, serum Na, serum osmolality, serum vasopressin or copeptin, food intake, etc, is really important even if the data has been published in other journals. It is sort of key to understanding the nature of the insult, especially given the multiple ways camels adapt to heat and dehydration.

All of these data, derived from the same animals used in this study, are presented in separate manuscripts and are appropriately cited in the text of this manuscript (references 24, 73, 74). We cannot republish these data! That said, for clarity, we have added the following text to the manuscript:

However, in comparing these data, we must consider the different protocols employed. Indeed, we note that the 72 hours WD protocol in that rat evokes a 2.5% increase in plasma osmolality²³, whereas the 20 day WD period in camels resulted in a more severe 13.5% increase in calculated serum osmolality²⁴. (page 9, lines 31-34)

The authors also do not describe (if I read the paper correctly) how the dehydration was done in the rats, and whether it was acute or chronic and again the severity of the dehydration using similar parameters. This is important from the standpoint of interpreting differences in response between species.

Again, we clearly refer in the text to our published papers from which these data were derived. That said, for clarity, we have added text in the results section that states that the rats were chronically water deprived for 72 hours (page 8, lines 15-16).

My guess is that this study is looking at both water and food deprivation in the camel, as spontaneous food intake will likely drop when no water is provided. This might be expected to activate AMPK and likely angiotensin II more as a mechanism to help maintain BP. Was BP measured? Perhaps this should be viewed as a limitation, that this is a study of both energy and water depletion.

We have added the following text to the discussion:

We note that it is a limitation of this study is that we did not measure cardiovascular parameters such as blood pressure and heart rate. (page 11, lines 25-26)

That said, it is a limitation of this study is that we did not measure food intake. (page 12, lines 3-4)

Finally, another limitation of the study should be mentioned, which is that most of the measurements are in RNA, and it does not necessarily reflect changes in protein.

A further limitations of this study is that we only studied steady-state transcript levels. Subsequent studies should address the proteomes of the SON and the posterior pituitary gland. (page 12, lines 11-13)

Finally, I think that the authors should also comment on the general histology (PAS or H/E) of the SON in the different states to better understand if there are histologic changes.

At the same time as the samples for this study were taken, tissue from the same animals was also harvested for morphological analysis. These studies, which include detailed electron microscopic analysis, are ongoing. A statement to this effect has been added to the Discussion:

Future studies will examine the effects of WD on the morphology of camel MCNs. (page 10, lines 21-22).

Note that when the hypothalamic samples for this study were collected, all other organs and tissues from these animals were taken. These samples are currently being studied by numerous groups around the globe. Thus, over the coming years, we will present a comprehensive picture of the overall physiological and molecular responses of the camel to dehydration and subsequent rehydration. This manuscript represents an important component of this overall endeavour.

All in all, however, the paper is very interesting and the authors are congratulated.

We thank the referee for his/her kind words.